# Isolation, Purification and Structure Identification of a Calcium-Binding Peptide from Sheep Bone Protein Hydrolysate

**DOI:** 10.3390/foods11172655

**Published:** 2022-09-01

**Authors:** Guanhua Hu, Debao Wang, Lina Sun, Rina Su, Mirco Corazzin, Xueying Sun, Lu Dou, Min Zhang, Lihua Zhao, Lin Su, Ye Jin

**Affiliations:** 1College of Food Science and Engineering, Inner Mongolia Agricultural University, Hohhot 010018, China; 2Agricultural and Animal Products Processing Research Institute, Inner Mongolia Academy of Agriculture and Animal Husbandry Academy, Hohhot 010018, China; 3Inner Mongolia Vocational College of Chemical Engineering, Hohhot 010010, China; 4Department of Agricultural, Food, Environmental and Animal Sciences, University of Udine, 33100 Udine, Italy

**Keywords:** sheep bone protein, calcium-binding peptide, characterization, separation, purification

## Abstract

To isolate a novel peptide with calcium-binding capacity, sheep bone protein was hydrolyzed sequentially using a dual-enzyme system (alcalase treatment following neutrase treatment) and investigated for its characteristics, separation, purification, and structure. The sheep bone protein hydrolysate (SBPH) was enriched in key amino acids such as Gly, Arg, Pro, Leu, Lys, Glu, Val, and Asp. The fluorescence spectra, circular dichroism spectra, and Fourier-transform infrared spectroscopy results showed that adding calcium ions decreased the α-helix and β-sheet content but significantly increased the random and β-turn content (*p* < 0.05). Carboxyl oxygen and amino nitrogen atoms of SBPH may participate in peptide–calcium binding. Scanning electron microscopy and energy dispersive spectrometry results showed that SBPH had strong calcium-chelating ability and that the peptide–calcium complex (SBPH–Ca) combined with calcium to form a spherical cluster structure. SBPH was separated and purified gradually by ultrafiltration, gel filtration chromatography, and reversed-phase high-performance liquid chromatography. Liquid chromatography-electrospray ionization/mass spectrometry identified the amino acid sequences as GPSGLPGERG (925.46 Da) and GAPGKDGVRG (912.48 Da), with calcium-binding capacities of 89.76 ± 0.19% and 88.26 ± 0.25%, respectively. The results of this study provide a scientific basis for the preparation of a new type of calcium supplement and high-value utilization of sheep bone.

## 1. Introduction

Calcium, which accounts for 1.5–2.2% of total body weight, is an essential micronutrient for human health [1,2]. Calcium deficiency is a public health problem all over the world, with the highest prevalence in children, pregnant women, middle-aged adults, and older adults [3]. Insufficient calcium intake can lead to a series of bone diseases, such as rickets and osteoporosis [4]. Although the serum calcium level can be maintained within the normal range through bone absorption, the calcium in food is an important supplement for the body’s bone calcium. People are accustomed to ingesting foods that contain a large amount of phytic acid, phosphoric acid, and oxalic acid. Thus, precipitates of insoluble phytate, phosphate, and oxalate with calcium ions form easily in the intestine, which reduces the solubility of calcium in the intestine and the bioavailability of calcium. Therefore, to avoid calcium deficiency, it is very important to take in adequate oral calcium at the appropriate time. Recent studies have found that peptides and proteins hydrolyzed from food are capable of chelating calcium and increasing its absorption and bioavailability in the form of a soluble Ca–peptide complex [5,6]. Peptide–calcium chelates have better solubility and thermal stability, which are essential characteristics for the persistence of calcium through food processing and calcium absorption in the human gastrointestinal system. Peptide–calcium complexes are promising alternatives to calcium-fortified functional foods or calcium supplements [7]. Thus far, researchers have isolated and identified an increasing number of calcium-binding peptides from various sources, such as DEEENDQVK from phosvitin [8]; GPAGPHGPVG, FDHIVY, and YQEPVIAPKL from tilapia bone collagen [9]; a novel calcium-binding peptide from chicken foot [10]; a peptide-purified form of wheat germ protein, FVDVT [11]; calcium-binding peptides from Antarctic krill protein hydrolysates, VLGYIQIR [12]; and so on. According to these findings, the calcium-binding capacity of the peptide is significantly increased compared to the capacity of the original protein hydrolysate following separation and purification.

China is the world’s largest producer of bone-derived food raw materials, and bone has considerable resource advantages. Sheep bone is a very nutritious natural resource that contains high-quality protein, reasonable proportions of fatty acids, rich minerals, and a variety of biologically active substances. However, because of the backward intensive processing technology of sheep bone, a lack of engineering technology and equipment, and a low degree of standardization, there are few high-value products and that leads to the underutilization of resources, serious waste of resources, and environmental pollution. Therefore, developing methods for carrying out intensive processing and high-value utilization of sheep bone resources and developing new, green, natural foods and dietary fortifiers has become an important research direction for comprehensive utilization of sheep bone resources.

In this study, calcium-binding peptides were obtained from sheep bone by enzymatic hydrolysis, and the structural properties of the peptides and peptide–calcium chelates were investigated. Subsequently, the calcium-chelating peptides were isolated and identified with Superdex peptide gel filtration chromatography, reversed-phase high-performance liquid chromatography (RP-HPLC), and liquid chromatography-electrospray ionization/mass spectrometry (LC/ESI-MS). The results of this study provide new directions for high-value utilization and development of sheep bone protein and provide a theoretical basis for sheep bone peptides as a functional component of calcium supplements.

## 2. Materials and Methods

### 2.1. Materials and Chemicals

Sheep bone was provided by Inner Mongolia Meiyangyang Food (Bayannaoer, China). Alcalase (>200,000 U/g), neutral protease (50,000 U/g), and flavor protease (30,000 U/g) were purchased from Beijing Soleibo Technology. HPLC-grade acetonitrile (ACN) and trifluoroacetic acid (TFA) were purchased from Tianjin Concord Technology (Tianjin, China). All other reagents were analytical grade.

### 2.2. Preparation of Sheep Bone Protein Hydrolysate (SBPH)

The residual meat, periosteum and bone marrow were thoroughly removed from the fresh bone, which was washed repeatedly in hot water to get rid of fat. The sheep bone was then boiled at 121 °C for 30 min in an autoclave, and after that rinsed with hot water to wipe off the grease, dried and ground into powder (80 order). The prepared bone powder was soaked for 24 h to remove fat in fluid changed every 6 h where the sample/anhydrous ethanol ratio was 1:10 (*w*/*v*); the residue was then washed with deionized water [13].

Defatted sheep bone meal was mixed with deionized water at a ratio of 1:20 (*w*/*v*). The substrate was initially hydrolyzed with the first enzyme for 2 h, followed by the second enzyme for 2 h, respectively, at the optimum temperature and pH, with an enzyme dosage of 12,000 u/g powder (6000 u/g: 6000 u/g). After proteolysis, the hydrolysates were heated to 100 °C for 15 min to inactivate enzymes and centrifuged at 8000× *g* for 20 min. The supernatant was lyophilized and collected at −20 °C.

### 2.3. Preparation of the Hydrolysate–Calcium Complex

The preparation of hydrolysate–calcium complex was performed as described by Sun et al. [14] with some modifications. Briefly, 2% (*w*/*v*) hydrolysate was added to 1% CaCl_2_ solution. The mixture was incubated in a water bath at 50 °C at a pH of 7.5 for 50 min. Upon completion of the chelation reaction, 9 times as much absolute ethanol was added to the reaction to remove the calcium. Then the complex was centrifuged at 8000× *g* for 15 min, and the precipitate was collected, lyophilized, and marked as SBPH–Ca. The calcium binding capacity was determined as Wang’s reported [11] with some modification. Next the bound calcium and total calcium content were measured with an atomic spectrophotometer (novAA 350^®^, Analytik Jena, Germany).
$$\mathrm{Cacium}-\mathrm{binding}\;\mathrm{capacity}\left(\%\right)=\frac{\mathrm{amount}\;\mathrm{of}\;\mathrm{chelated}\;\mathrm{calcium}\left(g\right)}{\mathrm{total}\;\mathrm{amount}\;\mathrm{of}\;\mathrm{calcium}\left(g\right)}\times 100\%$$

### 2.4. Amino Acid Analysis

The amino acid composition was determined with a high-speed amino acid analyzer (L-8900; Hitachi, Tokyo, Japan) according to the National Standard of the People’s Republic of China (GB 5009.124-2016, National Food Safety Standards, Determination of Amino Acids in Food).

### 2.5. Characterization of SBPH and SBPH–Ca

#### 2.5.1. Ultraviolet Spectroscopy

As described previously [15], the lyophilized SBPH sample was dissolved to 1 mg/mL in deionized water, and the pH was adjusted to 8.0. Then the SBPH–Ca by adding various concentrations of CaCl_2_ (20, 40, 60, and 80 mM) to a 1 mg/mL SBPH solution at 50 °C for 1 h. Absorbance was recorded over the wavelength range of 200 nm–400 nm with a TV-1810 spectrophotometer (Zhengzhou North–South Instrument Equipment). Deionized water was run as a blank calibration before measurement.

#### 2.5.2. Fluorescence Spectroscopy

This experiment refers to the research method of Beyer et al. with some modifications [16]. The fluorescence spectra of SBPH (0.1 mg/mL), containing different concentrations (40, 60, 80 mM) of CaCl_2_, were measured with an excitation wavelength of 295 nm and emission wavelengths ranging from 310 to 600 nm with an FLS1000 fluorescence spectrophotometer (FLS, Edinburgh, UK) after incubation at 37 °C for 30 min.

#### 2.5.3. Zeta Potential

The zeta potentials of SBPH and SBPH–Ca were analyzed with a Zetasizer Nano ZS90 particle size analyzer (Malvern Panalytical, Malvern, UK). The measurements were obtained as described in a previous study (S. Lin et al., 2017). A sample (1 mg/mL) was added to a U-shape cell and equilibrated at 25 °C for 60 s. All measurements were conducted at 25 °C and repeated 12 times.

#### 2.5.4. Scanning Electron Microscopy (SEM) and Energy Dispersive Spectroscopy (EDS)

The surface morphologies of SBPH and SBPH–Ca were observed with a scanning electron microscope (JSM-7800F; JEOL, Tokyo, Japan). The measurement was performed as described previously in method [17], where a freeze-dried powder sample was evenly coated on an SEM sample column with conductive adhesive, sprayed, and then sputter-coated with gold under a high vacuum condition. The scanning conditions were an acceleration voltage of 15.0 kV, a beam current of 6.9 × 10^−2^ mA, a working distance of 6.7 mm, and morphology observation at a magnification of 3000×. The element composition and content of the sample were detected with an energy dispersive spectrometer (S-4800; Hitachi, Tokyo, Japan) as described by Xue et al. [18].

#### 2.5.5. Circular Dichroism (CD) Spectra

SBPH and SBPH–Ca were prepared in deionized water at a concentration of 10 mg/mL and evaluated by Jasco J-815 system (Japan Spectroscopic Company, Tokyo, Japan) as described by Chang et al. [19]. The test conditions were a sample cell thickness of 0.1 cm, a bandwidth of 1.0 nm, a wavelength range of 190–320 nm, and a scanning speed of 100 nm/min. The average value of three scans was taken.

#### 2.5.6. Fourier-Transform Infrared Spectroscopy (FTIR)

Chemical interactions in SBPH and SBPH–Ca were determined using a FTIR spectroscopy [20]. Dry powder SBPH and SBPH–Ca samples were mixed with KBr and extruded onto a transparent sheet. FTIR spectra were recorded from 4000 to 400 cm^−1^ with an FTIR spectrometer (Bruker, Ettlingen, Germany).

#### 2.5.7. X-ray Diffraction (XRD)

The crystal structures of the peptide and peptide–calcium chelate were investigated with an X-ray diffractometer (Bruker Daltonic, Germany) with a Cu target anode material according to the method of Zhang et al. [21]. The system was operated at 2θ of 10–80°, 40 kV, 40 mA, with a scanning speed of 5°/min.

### 2.6. Isolation and Purification of Calcium-Binding Peptides

#### 2.6.1. Ultrafiltration

SBPH was filtered through ultrafiltration tubes (30, 10, and 3 KDa), concentrated by rotary evaporation, pre-frozen at −80 °C for 2 h, vacuum frozen for 24 h, and then stored at −20 °C for later use [22].

#### 2.6.2. Superdex Peptide Gel Filtration Chromatography

After ultrafiltration, the fraction with the highest calcium-binding capacity was collected and loaded to a Superdex 30 Increase 10/300 GL column (1 × 30 cm; Cytiva, Sweden) for further purification as described previously with a minor modification [11]. Distilled water with a flow rate of 0.5 mL/min was used as the eluent, and the fraction was collected with an automatic step collector (0.5 mL/tube). The mass concentration of the sample was 10 mg/mL, and the sample volume was 500 µL. Absorbance was measured at 280 nm with an AKTA liquid chromatography system (GE Healthcare Life Sciences, New York City, NY, USA), and each fraction was pooled and lyophilized for further determination.

#### 2.6.3. RP-HPLC

Active peptides with high calcium-chelating ability obtained by the AKTA protein purification instrument were weighed accurately, dissolved in distilled water to a 5 mg/mL solution, and filtered with a 0.22 µm aperture filter. Then 200 µL filtrate was injected into an InfinityLab Poroshell 120 EC-C18 column (4.6 × 150 mm, 4 µm; Agilent Technologies, Santa Clara, CA, USA) connected to an HPLC system (Agilent 1260). The operation was based on the method of the previous literature with some modification [9]. Elution was performed with solution A (0.05% TFA in water) and solution B (0.05% TFA in ACN), with a gradient of 0–5% B for 5 min, 30% B for 5–15 min, and 5% B for 15–20 min, at 1.0 mL/min with a UV detection wavelength of 280 nm at a column temperature of 35 °C. All fractions were collected and freeze-dried to measure calcium-binding activity.

#### 2.6.4. Identification of Peptides by Mass Spectrometry

The molecular mass and amino acid sequence of the purified peptide were determined with a Q Exactive Plus mass spectrometer (Thermo Scientific, Dionex Softron, Germany) with ESI over a range of 300–1800 *m*/*z*. Mass spectrometry data were retrieved with MaxQuant 1.6.1.0, and the P02845 protein database was downloaded from UniProt. Uniprot Ovis aries (Sheep) [9940]-63944-20210730. Fasta was used as the database.

#### 2.6.5. Peptide Synthesis and Validation of Calcium-Chelating Activity

A peptide with a purity greater than 98% was synthesized by Gill Biochemical (Shanghai, China), and its molecular mass was determined by LC/ESI-MS.

### 2.7. Statistical Analysis

All experiments were conducted at least three times, and the results are presented as means ± standard deviation (SD). Data were analyzed with SPSS (ver. 21.0; IBM). Differences among samples were analyzed by analysis of variance (ANOVA) with a significance cutoff of *p* < 0.05.

## 3. Results and Discussion

### 3.1. Characteristics of SBPH

Enzymatic hydrolysis can improve protein digestibility, can reduce allergic reactions, and is an effective way to produce high-function peptides. Each enzyme has a specific cleavage site. Dual-enzyme stepwise hydrolysis was used to obtain a protein hydrolysate with higher biological activity. Our previous research found that hydrolysates produced by alkaline protease, neutral protease, and flavor protease have higher calcium-binding capacities. Therefore, these three enzymes were selected for the dual-enzyme stepwise hydrolysis. As shown in Table 1, an initial enzymatic hydrolysis with alkaline protease followed by hydrolysis with neutral protease produced a higher result than other dual-enzyme hydrolysis combinations. Wu et al. produced a calcium-binding peptide from pig bone collagen using a combination of alcalase and neutral enzymes [23]. The combination of alkaline protease and neutral protease exposes the binding sites on the side chains of amino acid residues originally buried in the protein, such as –NH_2_, –COOH, and hydrophobic groups, and thus the calcium-binding ability of the enzymatic hydrolysate is improved.

The amino acid composition determines the biological activity of a peptide to a certain extent. Collagen is rich in Gly(G), Pro(P), and Hyp as well as bioactive peptides in its hydrolysate. The main component of sheep bone is collagen. As shown in Table 2, the Gly had the highest content in SBPH (15.68 ± 0.16%), followed by Ala, Arg, Pro, Leu, Lys, Glu, Val, and Asp; Met had the lowest content. In addition, the total content of Gly, Asp, Glu, Ser, Lys, Leu, and Arg was 45.42 ± 0.79%. These compounds are calcium-binding amino acids with high calcium-chelating activity [24]. A previous study successfully prepared egg white peptide–calcium chelates and found that Glu, Asp, Cys, Thr, Gly, and Lys play a meaningful role in the chelation process [25]. In addition, the negative charge and hydrophobic interaction of hydrophobic amino acids are favorable for chelation with calcium [26,27]. In this study, the hydrophobic amino acid content of SBPH was 36.26 ± 0.92%, which could improve the binding capacity of SBPH. Based on these results, SBPH appears to be a potential source of calcium-binding peptide.

### 3.2. Characterization of SBPH and SBPH–Ca

#### 3.2.1. Ultraviolet Absorption Spectroscopy

Protein absorbs ultraviolet light, mainly because of absorption by the side chain groups of tyrosine (Tyr), tryptophan (Trp), phenylalanine (Phe), His, and Cys residues as well as strong absorption by peptide bonds. As shown by the spectra in Figure 1, significant absorption peaks corresponding to a carbonyl group and aromatic amino acid residue were observed at about 226 nm and 268 nm. There was also an absorption peak near the ultraviolet region at 220 nm. As the concentration of calcium ions increased, this peak shifted to shorter wavelengths, and the absorbance of the absorption peak decreased from 2.719 to 2.445, which indicates a hypochromic effect and blue shift trend. This may have occurred because carbonyl oxygen and amino nitrogen in the peptide bond participated in the chelation reaction and generated new substances, which led to changes in the electrons of the amide bond and thus affected the ultraviolet absorption characteristics of the peptide. In addition, SBPH and SBPH–Ca showed a weak absorption peak near 260 nm, and the absorbance of this peak decreased from 0.737 to 0.404, possibly because of the transition caused by the π→π* electron in the ligand (N–C–O). A similar phenomenon was reported in a previous work [17], which demonstrated that the chirality of chromospheres (C=O and –COOH) and autochromes (–OH and –NH_2_) generated changes in polarization due to the peptide binding of calcium ions.

#### 3.2.2. Fluorescence Spectroscopy

Fluorescence spectroscopy can be used to explore interactions between organic ligand groups and metal ions by examining changes in wavelength and fluorescence intensity in spectr [28]. Some amino acid residues in protein or peptide molecules, such as Trp, Tyr, and Phe, have fluorescence properties and can effectively fluoresce at specific excitation wavelengths [29]. When calcium was added, the intensity of the endogenous fluorescence at approximately 330 nm increased dramatically from 3.27 × 10^5^ to 7.48 × 10^5^, and the absorption peak produced by Trp had red shifts from 335 to 357 nm (Figure 2), possibly because the positive charge of the residues near the phenyl end of Trp caused the peptide to produce red-shifted fluorescence in the fluorescence spectrum [17]. Similar results were obtained in previous studies [30,31]. However, different results indicating that the intensity of the endogenous fluorescence of the snapper fish scale protein hydrolysate–calcium complex decreased as the concentration of calcium ions increased have been reported [32]. This discrepancy could be due to complex interactions between chromophores and calcium, which result in changes in the energy of the excited state and thus the fluorescence intensity [33]. Hence, the results demonstrate that calcium ions chelated with peptides might cause the folding and aggregation of peptides [34].

#### 3.2.3. Zeta Potential

The surface charge of a substance is reflected by its zeta potential, a physicochemical parameter. The partial ionization of amino acid residues generates charges on the surface of a protein. As shown in Figure 3, compared to the SBPH, the zeta potential of theSBPH-Ca decreased significantly from −12.9 to −15.9 mV (*p* < 0.05). These data indicate that electron transfer occurs when SBPH react with Ca^2+^, resulting in new compounds, and that calcium ions can be chelated with positively charged groups, such as amino groups. These results show the same trend as previous research [35].

#### 3.2.4. SEM and EDS Analyses

The microstructures of SBPH and SBPH–Ca are shown in Figure 4. In the 3000× field of view, the surface of SBPH exhibited an irregular lamellar structure with a compact and smooth plate-like shape, whereas the surface of SBPH–Ca appeared loose and rough, with many pores of different sizes and clustered spherical particles. This distinction might be due to the folding and aggregation of the internal structure caused by the coordination bond between the peptide and calcium [36]. Moreover, a bridging effect was formed by the combination of amino and carboxyl groups in the peptide with calcium, which can also change surface properties [33]. This phenomenon is similar to cases of egg white peptide–calcium chelate [37] and chicken foot broth byproduct peptide–calcium chelate [10], which shows that chelation with calcium ions results in a more compact, granular, and crystal-like structure.

EDS was used to detect the basic element composition and content of SBPH and SBPH–Ca. As shown in the figure, the EDS spectra of SBPH showed prominent peaks for oxygen, sodium, sulfur, chlorine, and potassium; the spectra for SBPH–Ca showed significant peaks for carbon, oxygen, sodium, sulfur, chlorine, and calcium. There were three Ca signals of SBPH–Ca and a calcium weight percentage of 4.57%. These findings reveal that SBPH has high calcium-chelating ability between calcium ions.

#### 3.2.5. CD Analyses

Conformational variation in the secondary structure of proteins or peptides is accompanied by changes in the position and absorption of the circular secondary spectrum [38]. The CD spectra of SBPH and SBPH–Ca are depicted in Figure 5. After the peptide–calcium complex formed, the absorption peak of the peptide shifted from 201 nm to 198 nm. A large negative CD peak appeared at approximately 200 nm in the far ultraviolet region, and the peak intensity of the SBPH–Ca was lower than that of the SBPH, which suggests that the secondary structure of the peptide–calcium chelate changed during the formation process. The peptide formed an amphipathic helix and β-sheet for coiled-coil oligomerization. Upon calcium complexation, the β-turn and random structures increased markedly by 10.7% and 31.12%, respectively, along with a significant decrease in the α-helix and β-sheet structures. The combination of peptides and calcium ions may expose hydrophobic groups, leading to the decay of intramolecular hydrogen bonds and reducing α-helices [30]. SBPH–Ca became more freely coiled than SBPH. These data indicate that calcium ions induce changes in the spatial structure of the peptide. The binding sites of the peptide hidden in the β-sheet are exposed, causing a random coil structure to form. The peptide that interacts with calcium ions might be dominated by β-turn and random structures.

#### 3.2.6. FTIR Analyses

The characteristic changes in FTIR absorption peaks are used extensively to reflect the interaction between metal ions and organic ligands of peptides. As depicted in Figure 6, the amide A band of SBPH, which is associated with N–H stretching [39], was observed at 3365.05 cm^−1^. When the peptide–calcium chelate formed, the wavenumber moved to 3391.30 cm^−1^, which indicates that –NH participated in the chelation reaction [40]. The important vibrational mode of amides is the amide I band, located between 1700 and 1600 cm^−1^, which is primarily derived from the C=O stretching vibration of the peptide bond. The SBPH band situated at 1657 cm^−1^ shifted to 1654 cm^−1^ in SBPH–Ca. Furthermore, the peak at 1407.67 cm^−1^ belonging to –COO– and representing the symmetric stretching vibration of side chain –COOH [41] moved to the higher frequency of 1416.62 cm^−1^ after calcium ions were added, possibly because H was replaced with calcium. Meanwhile, the in-plane vibration of the O=C–N bond (633.6 cm^−1^) of SBPH shifted to 611.59 cm^−1^, possibly because of the increase in electron cloud density around –C=O in O=C–NH_2_ when combined with calcium [42]. Chen et al. reported that amino nitrogen atoms and oxygen atoms belonging to carboxylate groups can provide key binding sites for calcium ions [43]. These results suggest that Ca^2+^ binds to SBPH via interactions with amino N atoms and carboxyl O atoms, which is consistent with previous findings [21,44].

#### 3.2.7. XRD

XRD is commonly used to characterize the spatial structure of a substance, and it is suitable for characterizing the system change from an original random amorphous structure to an amorphous and crystalline doped structure from before to after the preparation of a chelate. The XRD spectra are shown in Figure 7. In the SBPH spectrum, seven main broad and weak dispersing diffraction peaks occurred near 14.78°, 25.92°, 29.7°, 31.7°, 42.2°, 49.16°, and 54.42°, respectively, which indicates that the peptide had a random amorphous structure (not an ordered structure). After binding with calcium, the five strongest and sharpest crystal diffraction peaks were at 22.68°, 23.58°, 25.6°, 31.9°, and 34.02°, with one at 37.82° and another at 46.2°, which reveals that the scattering intensity increased drastically after crosslinking with Ca^2+^ because of chain associations and the formation of junction zones. All these results imply that the combination of peptide and calcium produces new interaction forces in the spatial structure and that its morphology changes obviously, which leads to the appearance of new diffraction peaks. The peptide aggregation state changes gradually from amorphous to crystalline, and crystallinity is enhanced. These results are consistent with previous reports indicating that the peptide–calcium complex is a new type of crystalline substance different from peptides [11,21].

### 3.3. Ultrafiltration

The mineral-binding capacity of peptides is related to their molecular weight and length. Four kinds of peptide components with different molecular weights were obtained by ultrafiltration. The calcium−binding ability of the <3 kDa group was significantly higher than the abilities of the other peptides and SBPH (*p* < 0.05); this was followed by SBPH and 3–10 kDa, respectively (Figure 8). This result may be attributable to the types and quantities of amino acids in the different peptide components [33]. A previous study found that the <5 kDa group was most active after ultrafiltration of mung bean protein hydrolysate [45]. Caetano-Silva et al. found that metal complexes with low−molecular-weight peptides [45,46], and low-molecular-weight peptides are more stable during in vitro gastrointestinal digestion [47]. Thus, the <3 kDa type was chosen for further study.

### 3.4. Separation and Purification of Calcium-Chelating Peptide

The product obtained after protein enzymatic hydrolysis was a mixture of peptides and amino acids in which the amino acid composition, sequence, and peptide chain length of the polypeptides were relatively complex. To further improve the calcium-chelating capacity of the peptide obtained by enzymolysis, the complex enzymolysis products need to be separated and purified. Gel filtration chromatography is a useful technique for separating proteins according to their molecular weight. When a protein sample passes through gel particles with a porous network structure, protein molecules with high molecular weights cannot enter the gel particles and are eluted rapidly. When the molecular weight of the protein is lower, the time needed to elute into the gel is longer, and ultimately proteins with different molecular weights are separated. As shown in Figure 9, six fractions were isolated, and the calcium-binding capacity of each fraction was obtained. Of the fractions, F4 had the strongest calcium-binding ability (84.55 ± 1.65%) and was collected for further separation. A previous study reported that low-molecular-weight peptides are extremelybeneficial to calcium binding [48].

F4 was collected and separated further by RP-HPLC (Figure 10). Three peaks were separated according to differences in polarity. The calcium-binding activity of F4-3 was the highest at 86.76 ± 0.42%, which was significantly (*p* < 0.05) higher than other components, and then the amino acid sequence was further analyzed by LC-ESI/MS. 

### 3.5. Identification and Synthesis of the Calcium-Binding Peptides

For peptide identification, the purified F4-3 was subjected to LC-HRMS/MS. In total, 23 peptides were identified (Table 3), with lengths ranging from 6 to 15 amino acid residues and molecular weights ranging from 600 to 1400 Da. Four of the peptides with a purity of 98% were selected for synthesis, and their calcium-binding activity was verified (Table 4). The MS/MS spectra of the peptides GPSGLPGERG and GAPGKDGVRG were shown in Figure 11, and the calcium-binding capacity of GPSGLPGERG (925.46 Da) and GAPGKDGVRG (912.48 Da) were 89.76 ± 0.19% and 88.26 ± 0.25%, respectively, which was significantly (*p* < 0.05) higher than the rates of the other two peptides and SBPH. Note that GPSGLPGERG and GAPGKDGVRG, which matched well with the type I collagen alpha 1 chain, had three contiguous canonical GXY sequences that play a vital role in stabilizing intrachain and interchain hydrogen bonds in the collagen structure [32]. It is interesting that some other collagen peptides with a high affinity for calcium that can effectively promote the binding ability of calcium, such as DGPSGPK from tilapia bone [49], GDKGEGEAGER and GEKGEGGHR from Pacific cod skin [50] and GPAGPHGPPG from Alaska pollock skin [51], have been identified and determined to be similar to GPSGLPGERG and GAPGKDGVRG.

Different amino acid compositions and amino acid sequences influence biological activity [2]. For example, the amino acid compositions of Asp and Glu provide a good environment for calcium-binding peptides. A variety of calcium-binding peptides have been identified from different proteins, all of which contain Asp and Glu, such as NDEELNK isolated from sea cucumber egg [52], and from non-fish scale protein hydrolysate [35]. Asp, Glu, and Gly residues are major amino acids involved in calcium binding in porcine plasma-derived peptides [53]. It has been suggested that Glu, Arg, Asp, Gly, and hydrophobic amino acid are essential to the functioning of calcium-binding peptides [54]. Lys is the most common calcium-binding ligand. Zhang identified the novel calcium-binding peptide KGDPGLSPGK from Pacific cod bone gelatin hydrolysate and verified that Asp-3 and Lys-10 are calcium-binding sites. A novel calcium-binding peptide, VLPVPQK, identified from casein, effectively promotes the transport and absorption of calcium ions, and also contains Val, Leu, Pro, and Lys [55]. Calcium-binding peptides extracted from bovine serum have also been confirmed to contain Lys. In addition, Ser and Arg residues participate in calcium coordination. Caseinophosphopeptides (CPPs; Ser-Ser-Ser-Glu-Glu) can increase calcium absorption and were the first peptides found to have the ability to bind calcium ions [56,57]. Studies have reported that calcium-binding peptides derived from desalted duck egg white [58] and algae [59] also contain Ser. Another study found that the arginine carboxyl group of bradykinin is a metal-binding site [60]. Many peptides with high calcium-binding capacity, such as PLLR, LALGR, LSPLAGR, and YTSVLR, have been identified from walnut protein hydrolysates [61].

Therefore, these results suggest that Ser(S), Asp(D), Glu(E), Leu(L), Lys(K), and Arg(R) may contribute to the high calcium-binding properties of the two peptides identified in this study.

## 4. Conclusions

In this study, alkaline protease combined with neutral protease was used to prepare hydrolysis with calcium-binding ability from sheep bone protein. Derived from SBPH, SBPH–Ca is a new substance. Morphological analyses showed that the chelation of calcium ions with SBPH causes intramolecular and intermolecular folding and aggregation, resulting in the formation of spherical cluster-like structures. The formation of the SBPH–Ca complex affects the original spatial structure of the peptide, increasing the content of random and β-sheet structures. SBPH can chelate with calcium via carboxyl and amino groups. The amino acid sequences GPSGLPGERG (925.46 Da) and GAPGKDGVRG (912.48 Da) have calcium-binding capacities of 89.76 ± 0.19% and 88.26 ± 0.25%, respectively. This peptide has not been reported previously. Further studies are required to evaluate the complexation of the identified peptides with calcium ions and the mechanisms of absorption and transport in vivo of calcium–peptide complexes in the gastrointestinal tract. Sheep bone is a good choice for a calcium-binding peptide derived from food resources, and its application can be expanded to that of a calcium supplement and dietary fortifier.

## Figures and Tables

**Figure 1 foods-11-02655-f001:**
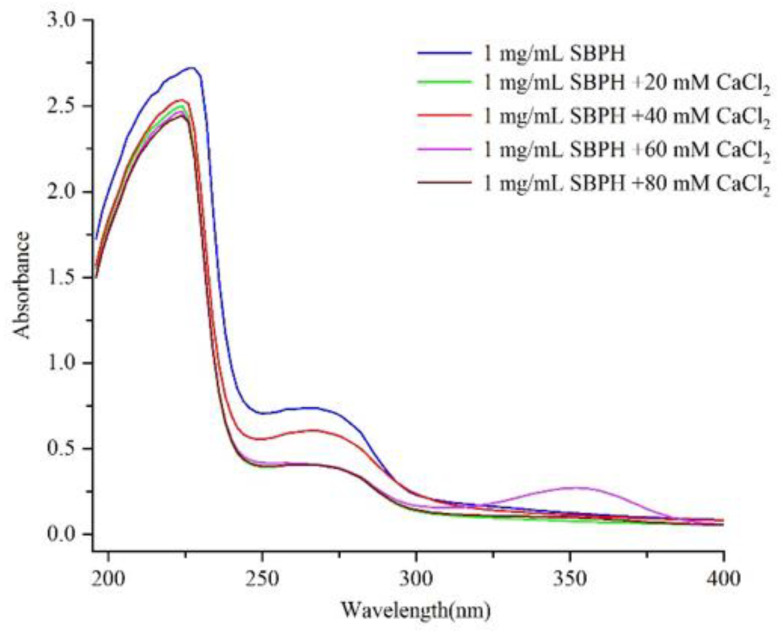
Ultraviolet-visible spectra of the SBPH with different CaCl_2_ concentrations within a wavelength range between 190 nm and 400 nm.

**Figure 2 foods-11-02655-f002:**
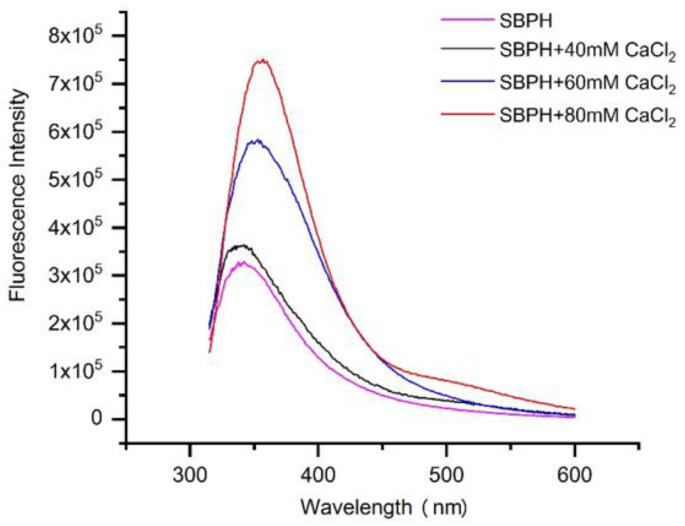
Fluorescence spectra of collagen peptide with different concentrations of CaCl_2_ over the emission wavelength ranging from 310 nm to 600 nm at an excitation wavelength of 295 nm.

**Figure 3 foods-11-02655-f003:**
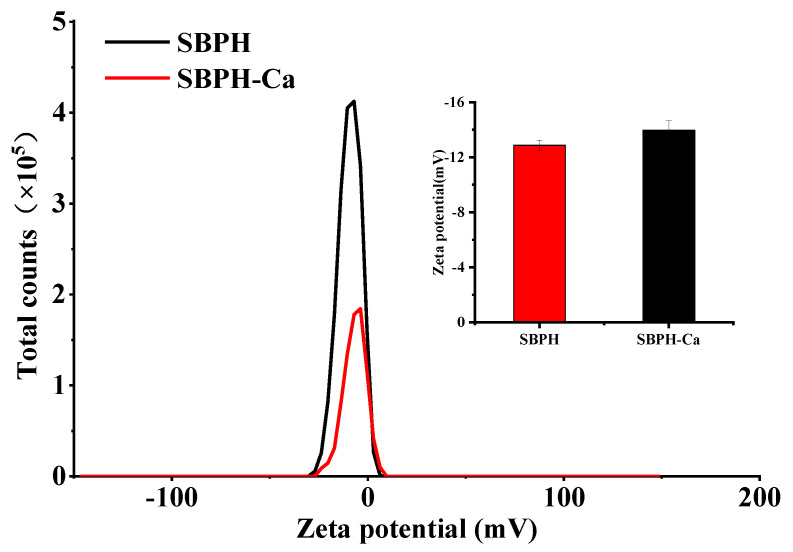
Zeta potential profile of the SBPH and SBPH−Ca by a Zetasizer Nano ZS90 particle size analyzer.

**Figure 4 foods-11-02655-f004:**
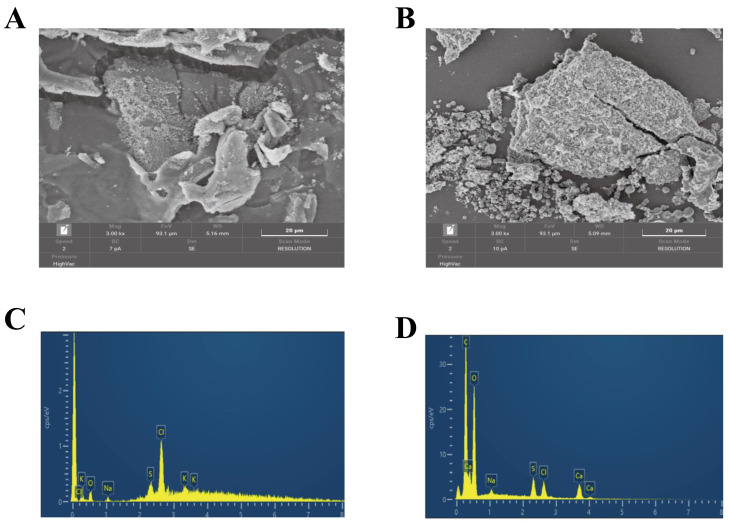
(**A**) SEM photograph of SBPH (3000). (**B**) SEM photograph of SBPH-Ca (3000). (**C**) EDS image of SBPH. (**D**) EDS image of SBPH-Ca.

**Figure 5 foods-11-02655-f005:**
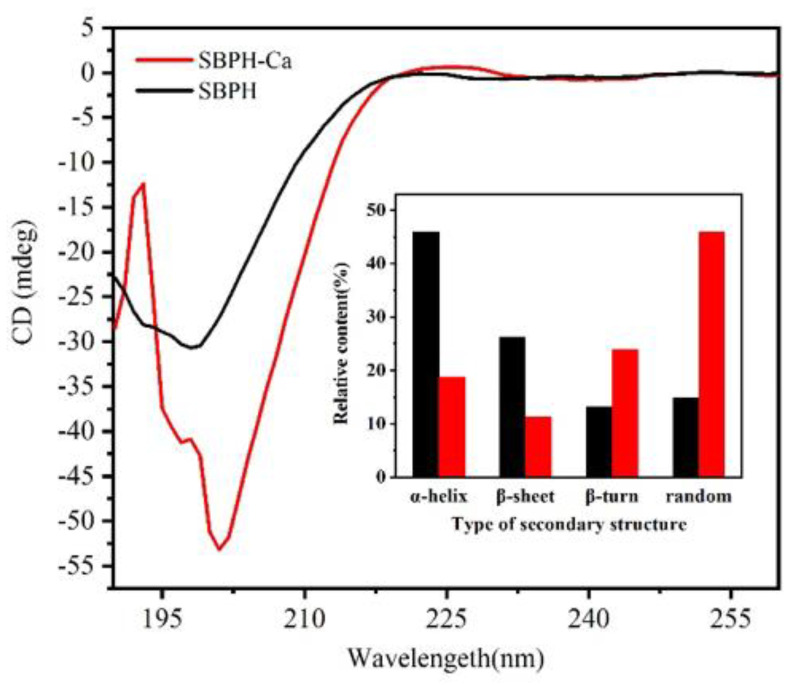
Far−UV circular dichroism (CD) spectroscopy of theSBPH and SBPH−Ca complex over the range of 190 nm to 260 nm, insets show their secondary structure relative content of α−helix, β−sheet, β−turn and random coil.

**Figure 6 foods-11-02655-f006:**
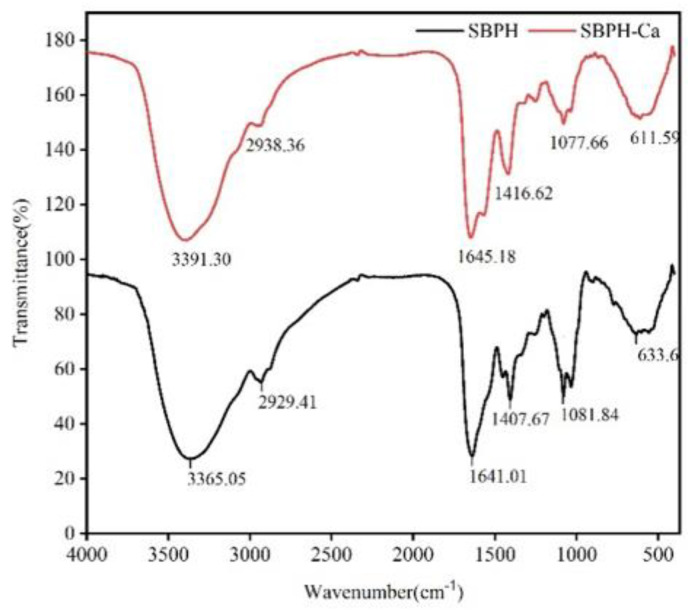
Fourier transform infrared spectra of the SBPH and SBPH−Ca within a wavenumber region between 4000 cm^−1^ and 400 cm^−1^.

**Figure 7 foods-11-02655-f007:**
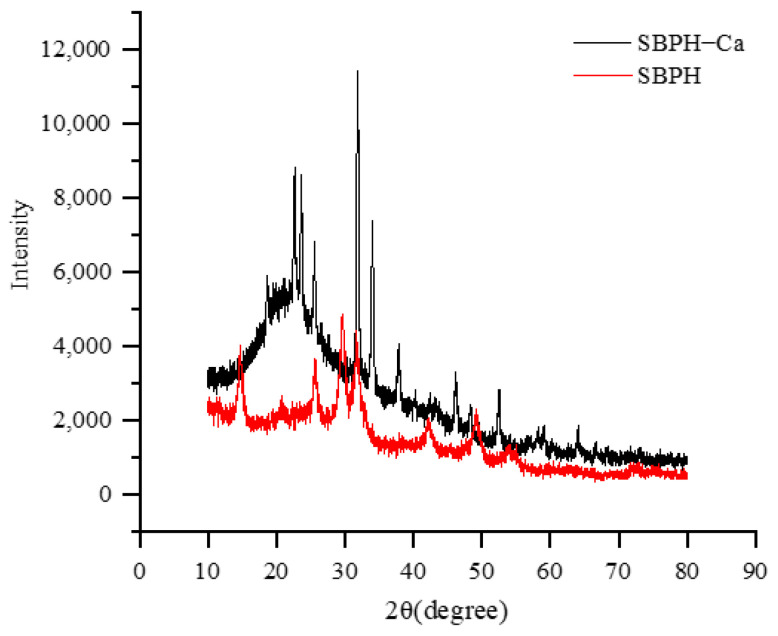
XRD patterns of the SBPH and SBPH−calcium complex.

**Figure 8 foods-11-02655-f008:**
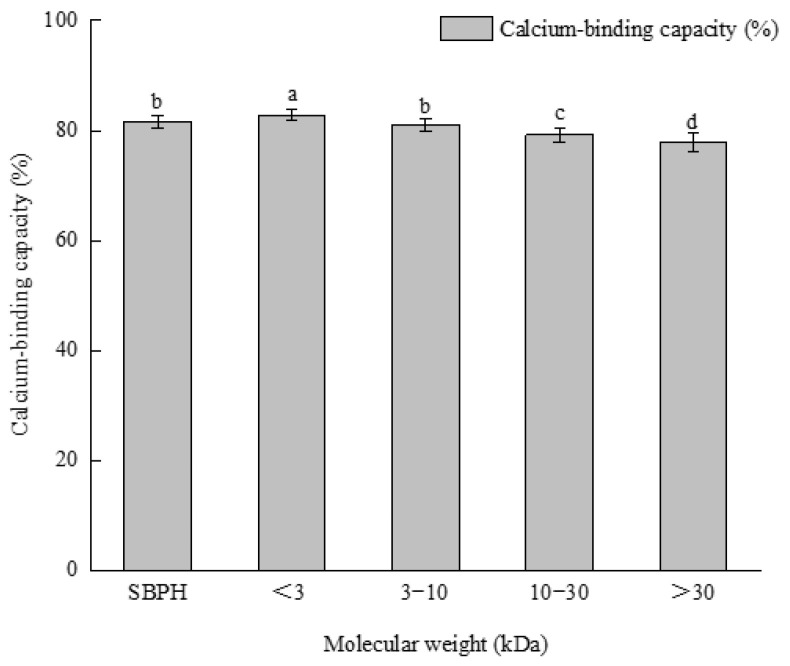
Calcium-binding capacity of SBPH as affected by the different molecular weight. Different letters represent significantly different values (*p* < 0.05). All measurements are expressed as means ± SD of three independent experiments.

**Figure 9 foods-11-02655-f009:**
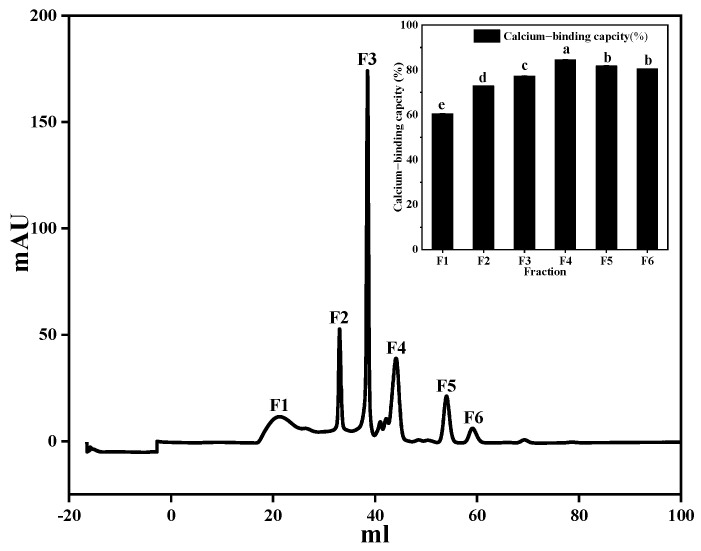
Fractionations (F1 to F6) separated from superdex peptide gel filtration chromatography and the calcium−binding capacities of F1 to F6. Different letters represent significantly different values (*p* < 0.05).

**Figure 10 foods-11-02655-f010:**
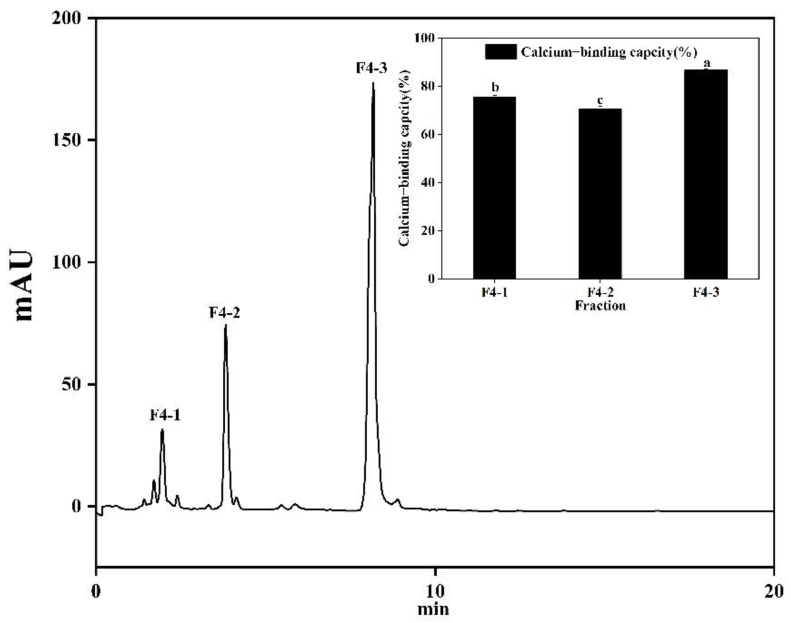
Fractionations separated from RP-HPLC and the calcium-binding capacities F4-1, F4-2, and F4-3. Different letters represent significantly different values (*p* < 0.05).

**Figure 11 foods-11-02655-f011:**
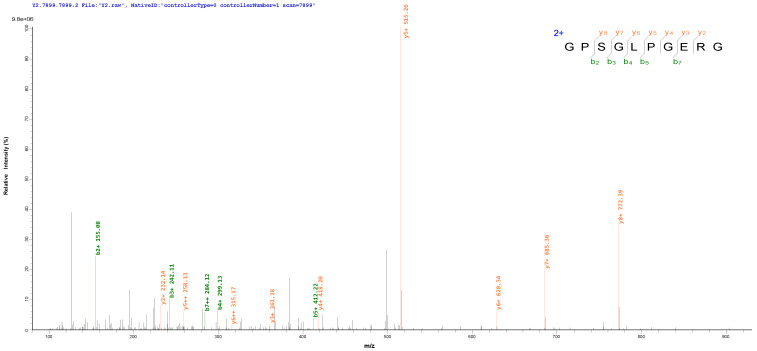
The MS/MS spectra of the peptides GPSGLPGERG and GAPGKDGVRG.

**Table 1 foods-11-02655-t001:** Comparison of collaboration treatment of alcalase, neutrase and flavor protease.

Groups	Time (h)	Temperature (°C)	pH	Protease (U/g)	Calcium Binding Capcity (%)
A→B	2→2	50	9→7.5	1:01	78.37 ± 0.22 ^b^
B→A	2→2	50	7.5→9	1:01	71.54 ± 0.09 ^d^
A→C	2→2	50	9→7.5	1:01	81.62 ± 0.32 ^a^
C→A	2→2	50	7.5→9	1:01	68.26 ± 0.18 ^e^
B→C	2→2	50	7.5→7.5	1:01	76.35 ± 0.25 ^c^
C→B	2→2	50	7.5→7.5	1:01	79.18 ± 0.12 ^b^

Note: A stands for alkaline protease; B stands for flavor protease; C stands for neutral protease; A→B represents A hydrolysis first and then B. The different letters mean significant differences *p* < 0.05 within the same indicator.

**Table 2 foods-11-02655-t002:** Amino acid composition of SBPH (g/100 g).

Amino Acid	Content	Amino Acid	Content
Asp	2.89 ± 0.05	Lys	5.04 ± 0.1
Thr	2.15 ± 0.02	NH_3_	2.27 ± 0.04
Ser	2.71 ± 0.06	His	2.34 ± 0.04
Glu	4.76 ± 0.09	Arg	8.77 ± 0.23
Gly	15.68 ± 0.16	Pro	6.49 ± 0.15
Ala	12.31 ± 0.52	a	85.21 ± 1.59
Val	4.11 ± 0.1	b	24.66 ± 0.38
Met	1.47 ± 0.08	c	36.26 ± 0.92
Ile	2.61 ± 0.05	d	16.14 ± 0.37
Leu	5.57 ± 0.13	e	7.65 ± 0.14
Tyr	2.33 ± 0.01	f	45.42 ± 0.79
Phe	3.71 ± 0.06		

Note. Data are denoted with mean ± standard deviation. a Total amino acids. b Essential amino-acid (sum of Thr, Ile, Leu, Val, Lys, Phe, Met). c Hydrophobic amino acid (sum of Ala, Val, Leu, Ile, Pro, Phe, Met). d Basic amino acid (sum of Lys, His, Arg). e Acidic amino acid (sum of Asp and Glu). f sum of Gly, Asp, Glu, Ser, Lys, Leu and Arg.

**Table 3 foods-11-02655-t003:** Identification of amino acid sequences of F4−3 fractions.

No.	Sequence	Length	Mass	Leading Razor Protein
1	HLDDLK	6	739.39	Q1A2D1 (76–81)
2	SDLSDL	6	648.30	Q28745 (82–87)
3	DLSDLH	6	698.32	Q28745 (83–88)
4	HLDDLP	6	708.34	Q28745 (73–78)
5	SDLSDLH	7	785.36	Q28745 (82–88)
6	HLDDLKG	7	796.41	Q1A2D1 (76–82]
7	SLVTGQT	7	704.37	W5Q638 (145–151)
8	DFGFDGD	7	771.27	W5NTT7 (1107–1113)
9	STGEIGPA	8	730.35	W5NTT7 (381–388)
10	FLPQPPQE	8	954.48	A0A6P7EK74 (1199–1206)
11	GEAGPQGPR	9	867.42	A0A6P7EK74 (352–360)
12	GAPGKDGVRG	10	912.48	A0A6P7EK74 (754–763)
13	VEGPPGPEGP	10	934.44	A0A6P7D8S3 (290–299)
14	IDGRPGPIGPA	11	1048.57	W5NTT7 (471–481)
15	GPAGPPGPIGN	11	932.47	A0A6P7EK74 (844–854)
16	SDGSVGPVGPA	11	941.45	W5NTT7 (234–244)
17	GADGAPGKDGV	11	942.44	A0A6P7EK74 (751–761)
18	DGAPGKDGVRG	11	1027.50	A0A6P7EK74 (753–763)
19	GIDGRPGPIGPA	12	1105.59	W5NTT7 (470–481)
20	GADGAPGKDGVRG	13	1155.56	A0A6P7EK74 (751–763)
21	DFLDEYIFLAVGR	13	1556.79	A0A6P3TAT2 (393–405)
22	GPEGPPGEPGPPGPP	15	1337.63	W5Q4M3 (1209–1223)
23	STGEAFVQFASQEIAEK	17	1840.88	A0A6P3YL92 (151–167)

**Table 4 foods-11-02655-t004:** Properties and Activities of Synthetic Peptides.

No.	Sequence	Mass	Organism	Protein Names	Description	Toxicity	Solubility	Allergenicity	Calcium-Binding Capacity (%)
1	GADGAPGKDGVR	1098.5418	Bos taurus (Bovine)	Collagen alpha-1(I) chain	Calcium, Metal-binding	NO	Good	NO	82.25 ± 0.11%
2	GPSGLPGERG	925.46174	Bos taurus (Bovine)	Collagen alpha-2(I) chain	Calcium, Metal-binding	NO	Good	NO	89.76 ± 0.19%
3	GAPGKDGVRG	912.47773	Bos taurus (Bovine)	Collagen alpha-1(I) chain	Calcium, Metal-binding	NO	Good	NO	88.26 ± 0.25%
4	AGPSGPSGLPGERG	1237.6051	Bos taurus (Bovine)	Collagen alpha-2(I) chain	Calcium, Metal-binding	NO	Good	NO	72.97 ± 0.44%

## Data Availability

Data is contained within the article.

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
