# Peer review of "Isolation, Purification and Structure Identification of a Calcium-Binding Peptide from Sheep Bone Protein Hydrolysate"

_foods, 2022, doi:10.3390/foods11172655_

Round 1

Reviewer 1 Report

1. Chapter 2: The procedure for obtaining defatted sheep bone is not described 2. Chapter 3 should be called Results and Discussion, since there is no separate chapter on the disk and the discussed section refers to the results of research by other authors 3. no statistical references in the text. The authors write about "significantly lower / higher values", but do not indicate the level of significance (although they indicate in the Material and Methods chapter that it is assumed to be p <0.05). I am not sure if it was just a translation error or if the statistical differences were really analyzed. There are several such provisions throughout the text. 4. in the notation "absorption peak of the peptide shifted from 201 to 198 nm" the unit nm should be added after 201 5. the authors write: "The combination of peptides and calcium ions may expose hydrophobic groups, leading to the decay of intramolecular hydrogen bonds and reducing α-helices. "Does this observation result directly from the research conducted? If not, there is no reference to the literature confirming this sentence. I assume that since there is no reference to the literature, these are hypotheses) not directly resulting from the research carried out. 6. references should be standardized (e.g. font type) in accordance with editorial requirements                      

Author Response

Response letter

Aug,20 2022

Dear Editors,

We would like to thank you and the reviewers’ for their constructive comments on our manuscript No: foods-1836884 entitled “Isolation, purification and Structure Identification of a Calcium-Binding Peptide from Sheep Bone Protein Hydrolysate. Those comments are all valuable and very helpful for revising and improving our paper”, as well as the important guiding significance to our future researches. We have read the comments carefully and revised accordingly. Any revisions to the manuscript have be marked up using the “Track Changes” function. Please see the revised manuscript and our point-by-point responses. We hope our revised version will now be acceptable for publication in Foods. Thank you for your time and consideration.Yours sincerely.Guanhua Hu

Response to Reviewer 1 Comments

Point 1: Chapter 2: The procedure for obtaining defatted sheep bone is not described 

Response 1: The procedure for obtaining defatted sheep bone had been described in 2.2 section(line97-line103).

Point 2:. Chapter 3 should be called Results and Discussion, since there is no separate chapter on the disk and the discussed section refers to the results of research by other authors

Response 2: Chapter 3 had been changed to results and discussion in line 213.

 Point 3: no statistical references in the text. The authors write about "significantly lower / higher values", but do not indicate the level of significance (although they indicate in the Material and Methods chapter that it is assumed to be p <0.05). I am not sure if it was just a translation error or if the statistical differences were really analyzed. There are several such provisions throughout the text. 

Response 3: Thank you for your suggestion, the level of significance had been supplemented in the corresponding sentence(line 316,447,452,464).

Point 4: in the notation "absorption peak of the peptide shifted from 201 to 198 nm" the unit nm should be added after 201

Response 4: Thank you for the observation, the unit nm had been added after 201 in line352.

 Point 5:  the authors write: "The combination of peptides and calcium ions may expose hydrophobic groups, leading to the decay of intramolecular hydrogen bonds and reducing α-helices. "Does this observation result directly from the research conducted? If not, there is no reference to the literature confirming this sentence. I assume that since there is no reference to the literature, these are hypotheses) not directly resulting from the research carried out. 

Response 5: The references had been listed to confirm this sentence in line 360.

Yang, X.; Yu, X.; Yagoub, A.-G.; Chen, L.; Wahia, H.; Osae, R.; Zhou, C. Structure and stability of low molecular weight collagen peptide (prepared from white carp skin) -calcium complex. Lwt 2021, 136, doi:10.1016/j.lwt.2020.110335.

Point 6: references should be standardized (e.g. font type) in accordance with editorial requirements       

Response 6: Thank you very much for your suggestion. References had been standardized in accordance with editorial requirements.

Reviewer 2 Report

The article reported the new peptides from sheep bone protein hydrolysates with calcium-binding capacities. It is suitable for the publication in Foods journal. Comments are listed as below.

1.      The first sentence of the abstract needs to be polished a bit. In scientific writing, it is inappropriate to use “we” as a subject. For example, line 21. Same problems are found in line 74, 113, and 476. Writing with a third-party perspective makes more sense.

2.      In the method and material section, the authors did not provide the references (citations) in every method used. Are they all new methods created by authors? Please correct this issue.

3.      In line 94 and 100 the centrifugation units seem not consistent. Please double check it.

4.      For calcium binding capacity, is it appropriate to express it as %. In method section, “Preparation of the hydrolysate–calcium complex”, the authors only stated 2% of hydrolysates in the assay. How could the authors relay the assay using hydrolysates to peptides (as shown in line 431-435)? How were the calcium binding capacity of the synthetic peptides?

5.      For SBPH, the authors should provide the information of protein contents, yield, peptides contents because they are important when it comes to commercialization.

6.      Some of the figures (Figure 9, 10 and 11) not clear. Please improve.

Author Response

Response letter

Aug,21 2022

Dear Editors,

We would like to thank you and the reviewers’ for their constructive comments on our manuscript No: foods-1836884 entitled “Isolation, purification and Structure Identification of a Calcium-Binding Peptide from Sheep Bone Protein Hydrolysate. Those comments are all valuable and very helpful for revising and improving our paper”, as well as the important guiding significance to our future researches. We have read the comments carefully and revised accordingly. Any revisions to the manuscript have be marked up using the “Track Changes” function. Please see the revised manuscript and our point-by-point responses. We hope our revised version will now be acceptable for publication in Foods. Thank you for your time and consideration.Yours sincerely.Guanhua Hu

Response to Reviewer 2 Comments

Point 1: The first sentence of the abstract needs to be polished a bit. In scientific writing, it is inappropriate to use “we” as a subject. For example, line 21. Same problems are found in line 74, 113, and 476. Writing with a third-party perspective makes more sense.

Response 1: Thank you very much for your suggestion. We had revised in the corresponding sentences(line 21,80,217,506).

Point 2: In the method and material section, the authors did not provide the references (citations) in every method used. Are they all new methods created by authors? Please correct this issue.

Response 2: In the method and material section, the references of each method had been supplemented.

Point 3: In line 94 and 100 the centrifugation units seem not consistent. Please double check it.

Response 3: Thank you for the observation, we apologize for the writing error. 8000 r/min had been revised to 8000*g in line 115.

Point 4: For calcium binding capacity, is it appropriate to express it as %. In method section, “Preparation of the hydrolysate–calcium complex”, the authors only stated 2% of hydrolysates in the assay. How could the authors relay the assay using hydrolysates to peptides (as shown in line 431-435)? How were the calcium binding capacity of the synthetic peptides?

Response 4:

(1)  For calcium binding capacity, is it appropriate to express it as %.

For calcium binding capacity, it is expressed in % and mg/g. We chose % to represent it referrring to the following reference.

Wu, W. ,  He, L. ,  Liang, Y. ,  Yue, L. ,  Peng, W. , &  Jin, G. , et al. (2019). Preparation process optimization of pig bone collagen peptide-calcium chelate using response surface methodology and its structural characterization and stability analysis. Food Chemistry, 284(JUN.30), 80-89.

Wang, L.; Ding, Y.Y.; Zhang, X.X.; Li, Y.F.; Wang, R.; Luo, X.H.; Li, Y.N.; Li, J.; Chen, Z.X. Isolation of a novel calcium-binding peptide from wheat germ protein hydrolysates and the prediction for its mechanism of combination. Food Chem. 2018, 239, 416-426, doi:10.1016/j.foodchem.2017.06.090.

Zhang, X. ,  Jia, Q. ,  Li, M. ,  Liu, H. , &  Liu, Z. . (2021). Isolation of a novel calcium-binding peptide from phosvitin hydrolysates and the study of its calcium chelation mechanism. Food Research International, 141(3), 110169.

Bao, Z.; Zhang, P .; Sun, N.;Lin, S. Elucidating the Calcium-Binding Site, Absorption Activities, and Thermal Stability of Egg White Peptide–Calcium Chelate.Foods 2021, 10, 2565. https://doi.org/

10.3390/foods10112565.

(2) In method section, “Preparation of the hydrolysate–calcium complex”, the authors only stated 2% of hydrolysates in the assay.

2% hydrolysate was added to 1% CaCl2 solution, which represents the addition ratio of hydrolysate and CaCl2 is 2:1.

(3) How could the authors relay the assay using hydrolysates to peptides (as shown in line 431-435)?

Thanks for your observation, we had revised in sections 3.2.3 (line 315-317) and 3.2.5(line 353-354).

(4) How were the calcium binding capacity of the synthetic peptides?

The calcium-binding capacity of synthetic peptides is higher than hydrolyzates, as described in section 3.5.

Point 5: For SBPH, the authors should provide the information of protein contents, yield, peptides contents because they are important when it comes to commercialization.

Response 5: Thank you very much for this suggestion, we have not considered this issue enough, but we will add it in future research directions.

Point 6: Some of the figures (Figure 9, 10 and 11) not clear. Please improve.

Response 6: The figures had been modified.

Round 2

Reviewer 2 Report

None.